# Thermophysical Properties of Nanofluid in Two-Phase Fluid Flow through a Porous Rectangular Medium for Enhanced Oil Recovery

**DOI:** 10.3390/nano12061011

**Published:** 2022-03-18

**Authors:** Abdullah Al-Yaari, Dennis Ling Chuan Ching, Hamzah Sakidin, Mohana Sundaram Muthuvalu, Mudasar Zafar, Yousif Alyousifi, Anwar Ameen Hezam Saeed, Muhammad Roil Bilad

**Affiliations:** 1Department of Fundamental and Applied Sciences, Universiti Teknologi PETRONAS, Seri Iskandar 32610, Malaysia; dennis.ling@utp.edu.my (D.L.C.C.); mohana.muthuvalu@utp.edu.my (M.S.M.); mudasar_20000296@utp.edu.my (M.Z.); 2Department of Mathematics, Faculty of Applied Science, Thamar University, Dhamar 00967, Yemen; p90499@siswa.ukm.edu.my; 3Department of Mathematical Sciences, Faculty of Science and Technology, Universiti Kebangsaan Malaysia, Bangi 43600, Malaysia; 4Department of Chemical Engineering, Universiti Teknologi PETRONAS, Seri Iskandar 32610, Malaysia; anwar_17006829@utp.edu.my; 5Faculty of Integrated Technologies, Universiti Brunei Darussalam, Gadong, Bandar Seri Begawan BE1410, Brunei; roil.bilad@ubd.edu.bn

**Keywords:** nanofluid injection, flooding, porous media, enhanced oil recovery, mathematical model

## Abstract

It is necessary to sustain energy from an external reservoir or employ advanced technologies to enhance oil recovery. A greater volume of oil may be recovered by employing nanofluid flooding. In this study, we investigated oil extraction in a two-phase incompressible fluid in a two-dimensional rectangular porous homogenous area filled with oil and having no capillary pressure. The governing equations that were derived from Darcy’s law and the mass conservation law were solved using the finite element method. Compared to earlier research, a more efficient numerical model is proposed here. The proposed model allows for the cost-effective study of heating-based inlet fluid in enhanced oil recovery (EOR) and uses the empirical correlations of the nanofluid thermophysical properties on the relative permeability equations of the nanofluid and oil, so it is more accurate than other models to determine the higher recovery factor of one nanoparticle compared to other nanoparticles. Next, the effect of nanoparticle volume fraction on flooding was evaluated. EOR via nanofluid flooding processes and the effect of the intake temperatures (300 and 350 K) were also simulated by comparing three nanoparticles: ${\mathrm{SiO}}_{2}$, ${\mathrm{Al}}_{2}O_{3}$, and CuO. The results show that adding nanoparticles (<5 v%) to a base fluid enhanced the oil recovery by more than 20%. Increasing the inlet temperature enhanced the oil recovery due to changes in viscosity and density of oil. Increasing the relative permeability of nanofluid while simultaneously reducing the relative permeability of oil due to the presence of nanoparticles was the primary reason for EOR.

## 1. Introduction

New technologies are being developed to boost oil production to comply with rising energy demand [1], while continuously depleting resources. Oil recovery processes are utilized to obtain maximum oil from hydrocarbon reservoirs [2] under economic feasibility constraints. Primary recovery refers to extracting oil from a reservoir using natural energy and pressure. Because the inherent energy of the reservoir declines throughout the production process, only 5% to 15% of hydrocarbons may be recovered by using this recovery method.

Enhanced oil recovery (EOR) is implemented to improve residual oil extraction from the homogenous medium in the reservoir [3]. The secondary and the tertiary recoveries are included in the EOR process, typically constituting sophisticated technologies. The secondary EOR process is achieved through water injection or gas injection [4]. After primary recovery, fluid injection is used to restore the reservoir pressure. In contrast, the fluid injection in the tertiary recovery alters the interaction between the reservoir rock and the fluid [5]. The displacement of oil from porous media using water or solutions including surfactants or polymer additives is a popular method of extracting oil from terrigenous sources [6]. 

Silicon ${\mathrm{SiO}}_{2}$, aluminum oxide ${\mathrm{Al}}_{2}O_{3}$, and copper oxide CuO nanoparticles are the most frequently utilized nanoparticles in EOR. As a result of their characteristics, nanoparticles can address a few issues with conventional approaches and improve the displacing fluid’s sweep efficiency, because the base fluids’ thermal conductivity and interfacial characteristics are both improved by nanoparticles. Furthermore, they can withstand high temperatures and pressures, are ecologically friendly and are cost-effective. Almost all EOR approaches are expensive, but the advantages of nanofluid flooding can be more economical. In addition, nanoparticles can optimize the properties of displacing fluids such as density, viscosity, the interfacial tension between oil and water, thermal conductivity, and specific heat [2].

A few EOR approaches involving nanomaterials have been reported recently. Fluids containing suspended nanoparticles have the potential to significantly enhance oil recovery procedures and increase oil extraction from reservoirs [7,8,9]. The incorporation of nanoparticles into the base fluid would affect the heat transport [10,11,12,13], the fluid viscosity [14,15,16,17,18,19,20], the thermal conductivity [21,22,23], and the rheological conductivity [22,24,25]. This approach is regarded as one of the alternate methods for locating new hydrocarbon reserves [26]. 

Several papers have recently been published in which nanofluid flooding is proposed as an agent for EOR [27,28,29,30,31]. The effect of nanofluid flooding on enhancing oil extraction from porous media has also been reported. Despite several experimental and modeling investigations, there is currently no exact explanation for the impacts of nanoparticles on the rise in oil recovery during nanofluid flooding, which limits the adoption of the method in the industry. Some suggested theories for understanding the influence of nanofluids on oil extraction include surface tension-lowering techniques [32,33,34,35], the alteration of the porous medium’s wettability [36,37,38], and the change in viscosity [39,40,41,42,43,44]. 

Researchers have looked at the transfer of microscopic particles in a two-phase flow through an open porous medium to simulate the EOR process [6,45,46,47,48]. Polysilicon nanoparticles were studied by Ju and Fan [49]. Their results showed that the environment shifts from oily to watery, promoting oil extraction. They developed a mathematical model for small particle transport in a two-phase flow in porous media [50]. 

The presence of nanoparticles also has a considerable impact on the thermophysical properties of the base fluid [51,52,53,54,55]. However, a limited number of research studies are available on thermophysical properties of nanofluid in porous media for EOR [54]. Furthermore, nanoparticles seem to enhance the effective characteristics of the injected fluid, particularly in the process of washing the oil from a porous medium [56,57].

In the present study, a numerical method of finite element is used to solve the governing equations that were derived from Darcy’s law and the mass conservation law of a two-phase incompressible fluid in a two-dimensional porous homogenous area. The thermophysical properties of nanoparticles were studied for their volume fraction and their kind and influence on the extraction of oil. Compared to earlier research, a more efficient numerical model is proposed. The proposed model allows for the cost-effective study of heating-based inlet fluid in EOR and uses the empirical correlations of the nanofluid thermophysical properties on the relative permeability equations of the nanofluid and oil, so it is more accurate than other models to determine the higher recovery factor of one nanoparticle compared to other nanoparticles. Moreover, the EOR via nanofluid flooding processes and the effect of the intake temperatures (300 and 350 K) were also simulated by comparing three nanoparticles, ${\mathrm{SiO}}_{2}$, ${\mathrm{Al}}_{2}O_{3}$ and CuO, with different nanoparticle volume fractions. Finally, we studied how increasing the inlet temperature enhanced the oil recovery due to changes in viscosity and density of oil. This work aims to model the EOR procedure by flooding, taking into consideration the efficient characteristics of nanofluids conveyed through a model porous medium.

## 2. Numerical Implementation

Computational fluid dynamics (CFDs) is a methodology used to examine systems, such as fluid flow and the heat transport system, through computer simulations. When compared to experimental investigations, the CFD technique has the potential to study critical and unique circumstances in a process, reduce response time and research costs, and gain a complete and detailed understanding of the process [58,59,60,61,62,63,64]. 

FEM techniques, which are one of the most used approaches for CFD research, were employed here. The primary framework of the employed simulation software COMSOL Multiphysics is fully based on FEM as a numerical method. The numerical analysis community frequently uses finite element methods to investigate numerical approaches for fluid flow. 

There is a lot of work on CFD and finite element techniques in literature, particularly in the EOR process and porous medium modeling [54,65,66,67,68]. Four phenomena occur when nanofluid is introduced into porous media: adsorption, desorption, blockage, and migration with flowing fluid. This section introduces one proposed model that others have recently adopted for nanofluid injection in oil reservoirs for EOR [69,70,71,72]. 

### 2.1. Model Assumptions

The following assumptions were used to build the two-phase flow mathematical model with distributed nanoparticles [73]:Fluids were both incompressible.Fluid flow in pore medium obeyed Darcy’s law.The flow was in an isothermal state.There was no capillary pressure such that permeability and porosity were independent of time and space and that the impact of gravity was neglected.Darcy’s law and the mass conservation law provided the essential equations. Mass conservation was comparable to volume conservation since the two phases are incompressible.

### 2.2. Geometry Creation

The geometry of the model was produced using COMSOL Multiphysics software with the following attributes: a 2D porous medium with porosity of 0.3, absolute permeability of $1e-9{\;m}^{2}$, and size of 1×2 $m^{2}$. Figure 1 indicates that the porous medium was homogenous for its distribution. 

The boundary layer mesh is important because the default triangular mesh in COMSOL software does not work well for fluid flow problems, and there is a boundary layer along the contact between the nanofluid and the solid, as shown in Figure 2a. A no-slip condition from our assumption generated the boundary layer of the nanofluid flow through a stationary porous medium, and the layer close to the solid has zero velocity. Figure 2b below depicts the development of a boundary layer mesh for achieving a refined solution to the fluid flow problems by using extremely small, thin components near the solid wall.

### 2.3. Grid Independency

Varying grid types with different mesh sizes were developed for the investigation of grid independence. To compare grids, fluid pressure drops in the porous medium were considered. The total pressure at the entrance and exit were estimated. The mesh size, number of elements generated, as well as the pressure drop between the intake and exit are all shown in Table 1. As a result, mesh number 3 is chosen as the ideal mesh number and is utilized in all the simulations that follow.

Although COMSOL Multiphysics meshing algorithms attempt to eliminate low-quality parts, this is not always achievable for all geometries. A quality of 1 is the greatest feasible value for any quality measure and signifies an ideal element within the specified quality measure. At the interval’s opposite end, 0 denotes a degraded element. As a result of the element measurements, the meshing of the present research geometry is of excellent quality, with a quality of 1, as shown in Figure 3a.

In addition, Figure 3b depicts the plot of convergence, which is time–domain convergence for time-dependent problems (variables), and finds a solution to the system of equations with error less than  1e−5 at a specific time, 250 s.

### 2.4. Governing Equations

The equations associated with the mathematical modeling of flooding problems using nanofluids accounted for the heat transfer. The nanofluid flooding equations were first studied as water flooding equations by assuming that the nanoparticle’s concentration equals 0 for examining the mathematical and physical equations of the problems in the oil reservoir. The porous medium’s heat equation was then supplied to accomplish the model. 

Because the initial pressure difference between the well and the tank was extremely large, the oil exited the tank automatically until the well and tank pressures equalized, at which time recovery procedures were required. Because it was considered that flow cannot exit boundaries 2 and 3, and that flow can only move between boundaries 1 and 4, the zero-flow boundary condition was considered.

In recent years, nanofluids have been used instead of water to increase oil recovery and to improve the flooding function [26]. To determine the rate of oil recovery, the equations of saturated oil ($S_{o}$) and water-based nanofluid saturation ($S_{\mathrm{nf}}$) in the environment should be determined. The porous medium was first filled with oil. Water-based nanofluid entered the porous medium from boundary 1 with a velocity $u_{\mathrm{nf}0}$ of 0.001 m/s, and displaced oil, which was permitted to flow out of the porous domain via boundary 4. At the inlet (boundary 1), the velocity $u_{o}$ of oil was 0: (1)$$-n.\rho_{o}u_{o}=0\;m/s$$
(2)$$-n.\rho_{\mathrm{nf}}\;u_{\mathrm{nf}}=0.001\;m/s$$

The pressure at the outlet (boundary 4) was set to 0 Pa:(3)$$p=p_{\mathrm{nf}}=0\;\mathrm{Pa}\;$$

The total velocity $u=u_{\mathrm{nf}}+u_{o}$ was constant in time and space because of the assumptions and boundary conditions stated above, and Darcy’s law and the mass conservation law provide the essential equations. Mass conservation is comparable to volume conservation since the two phases were assumed to be incompressible.

For the two phases of water-based nanofluid and oil, Darcy’s law yields [74]:(4)$$u_{\mathrm{nf}}=-Km_{\mathrm{nf}}\;\nabla p$$
and
(5)$$u_{o}=-Km_{o}\nabla p$$
where the velocities of phase 1 water-based nanofluid and phase 2 oil are $u_{\mathrm{nf}}$ and $u_{o}$, respectively; the mobilities of phases 1 and 2 are $m_{\mathrm{nf}}$ and $m_{o}$, respectively; $m_{\mathrm{nf}}=k_{\mathrm{rnf}}/\mu_{\mathrm{nf}}$; and $m_{o}=k_{\mathrm{ro}}/\mu_{o}$. The functions $k_{\mathrm{nf}}$ and $k_{o}$ are dependent on the saturation, and K is the absolute permeability.

The laws of volume conservation are:(6)$$\nabla .\left(\rho_{\mathrm{nf}\;}u_{\mathrm{nf}}\right)=-{\varepsilon \;\rho }_{\mathrm{nf}\;}\frac{\partial S_{\mathrm{nf}}}{\partial t}$$
and
(7)$$\nabla .\left(\rho_{o\;}u_{o}\right)=-{\varepsilon \;\rho }_{o\;}\frac{\partial S_{o}}{\partial t}$$
where the saturation of phase 1 (nanofluid) is $S_{\mathrm{nf}}$ and of phase 2 (oil) is $S_{o}$, while the porosity is ε. The initial condition was taken as $S_{\mathrm{nf}}=0$ to solve saturation equations. In the reservoir, there was often not just oil, but also some water. In this study, for first saturation, a quantity of 0 was considered. Because the input fluid contains no oil, the inlet nanofluid saturation ($S_{\mathrm{nf}}$) was 1. It should also be noted that there is a correlation between the amount of oil saturation and nanofluid flooding [75]:(8)$$S_{\mathrm{nf}}+S_{o}=1$$

Equation (9) was obtained by adding Equations (6) and (7) and by considering the fluids incompressible, and by using Equation (8).
(9)$$\nabla .\left(u_{\mathrm{nf}}+u_{o}\right)=0$$
and
(10)$$u=u_{\mathrm{nf}}+u_{o}$$

The entire volumetric flow (velocity) is denoted by u. In this study, we simplify by assuming that u is constant.

Equations (4) and (5) are added together, then Equation (10) is used to produce Equation (11).
(11)$$u=-K\left(m_{\mathrm{nf}}+m_{o}\right)\nabla p$$

Solving Equation (11) for ∇p and plugging it into Equation (4) led to Equation (12).
(12)$$u_{\mathrm{nf}}=\left(m_{\mathrm{nf}}\;u\right)/\left(m_{\mathrm{nf}}+m_{o}\right)$$

Substitution of Equation (12) for Equation (6) resulted in Equation (13).
(13)$$\nabla .\left(\rho_{\mathrm{nf}\;}\left(\left(m_{\mathrm{nf}}\;u\right)/\left(m_{\mathrm{nf}}+m_{o}\right)\right)\right)=-{\varepsilon \;\rho }_{\mathrm{nf}\;}\frac{\partial S_{\mathrm{nf}}}{\partial t}$$

Since water-based nanofluid is liquid, the values of effective density and viscosity of nanofluid to solve Equation (13) should be utilized. Equations (14) and (15) were used to derive the effective density and viscosity, respectively [76]:(14)$$\rho_{\mathrm{nf}}=\emptyset \rho_{\mathrm{np}}+\left(1-\emptyset \right)\rho_{w}$$
(15)$$\mu_{\mathrm{nf}}=\mu_{w}\left(1+39.11\emptyset +533.9\emptyset^{2}\right)$$
where $\rho_{\mathrm{nf}}$ and $\mu_{\mathrm{nf}}$ are the density and viscosity of nanofluid, respectively, and ∅ is the nanoparticle volume fraction of the nanofluid. 

Relative permeability was required for water-based nanofluid and oil for solving the saturation equations. The effective saturation ($S_{e}$) was utilized in the relative permeabilities:(16)$$S_{e}=\frac{S_{\mathrm{nf}}-S_{\mathrm{rnf}}}{1-S_{\mathrm{rnf}}-S_{\mathrm{ro}}}$$
where ($S_{e}$) is the effective saturation, ($S_{\mathrm{rnf}}$) is the residual nanofluid saturation, and ($S_{\mathrm{ro}}$) is the residual oil saturation. Equation (17) [77] defines the relative permeability of water-based nanofluid and oil in the Brooks–Corey model:(17)$$k_{\mathrm{rnf}}=S_{e}3+\frac{2}{\lambda },\;k_{\mathrm{ro}}=\left(1-S_{e}^{2}\right)\times \left(1-S_{e}^{1+\frac{2}{\lambda }}\right)$$
where λ is the distribution index for the pore size, and its value is 1. The capillary pressure or the difference in pressure between nanofluid and oil equals 0 since it was assumed that there is no capillary pressure in this model. It can then be expressed as follows.
(18)$$0=P_{c}=P_{o}-P_{\mathrm{nf}}$$
then
(19)$$p_{o}=p_{\mathrm{nf}}=p$$

The oil temperature in the reservoir increased at a higher heat transfer rate between the two phases when the intake fluid was high. When nanoparticles were added, the thermal conductivity coefficient $\left(k_{eff}\right)$ increased, allowing for more efficient heat transmission involving nanofluids and oil. Under this situation, the flooding approach would benefit from two factors as temperatures rise. Lowering the viscosity of the oil eased the nanofluid for replacing the oil from the porous medium since its stickiness and heavyweight were reduced, and it drained faster than the same water flow. Another effect of rising temperatures was that oil density decreased, making it lighter in volume and needing a smaller amount of energy to be evacuated from reservoirs. 

The energy equation in porous media is introduced in this section [78]:(20)$${\left(\rho C_{p}\right)}_{eff}\frac{\partial T}{\partial t}+\rho C_{p}u.\nabla T+\nabla .q=0,\;q=-k_{eff}\nabla T$$
(21)$${\left(\rho C_{p}\right)}_{eff}=\theta_{p}\rho_{p}C_{p,p}+\left(1-\theta_{p}\right)\rho C_{p},\;k_{eff}=\theta_{p}k_{p}+\left(1-\theta_{p}\right)k$$

In the environment, the initial temperature is 300 K. The border condition is specified as Equation (22) for the input of the heat equation:(22)$$-nq=\rho ∆Hun,\;∆H=\int_{T_{in}}^{T}C_{p}dT$$

The target variable for this study is $T_{in}$, denoting the temperature of the input fluid and influencing the performance of the recovery process. The condition of the foreign border 4 is:(23)−nq=0

Table 2 illustrates the properties of water at a constant temperature. The four fluid parameters used in the two-phase and energy equations are density, viscosity, specific heat capacity, and conduction heat transfer coefficient. It also summarizes the results of past studies on oil characteristics. With rising temperatures, the density and viscosity of water and oil change. Transferring water heat to oil decreases the viscosity and density of oil. Therefore, the oil could be transported more readily and recovered. This approach must be investigated in this study. According to Equation (24), the density of water varies with temperature (for temperatures ranging from 300 K to 350 K) [79]. Table 3 also shows how the viscosity of water changes with temperature.
(24)$$\rho \left(T\right)=838.46613+1.400506T^{1}-0.00301123T^{2}+3.718e-7T^{3}$$

After assessing how temperature affects the density and viscosity of water, the mechanism of temperature in affecting the density and viscosity of the oil was then evaluated. There are data about oil density at three temperatures. Oil characteristics, like water, are temperature-dependent. The change in oil density with temperature is shown in Table 4. Similarly, oil viscosity operates in the same way. Table 5 [82] shows the oil’s viscosity at three distinct temperatures.

As the oil temperature inside the reservoir rises, the viscosity and the density of the oil gradually reduce, as indicated in Table 4 and Table 5. The oil becomes smoother and easier to evacuate from the porous medium. Therefore, lowering these parameters would also aid in EOR. 

### 2.5. The Nano-Particles’ Effect on Thermophysical Properties of Nanofluid

The nanoparticles ${\mathrm{SiO}}_{2}$, ${\mathrm{Al}}_{2}O_{3}$, and CuO were evaluated in this study. These nanoparticles were combined with water at specific compositions to generate a nanofluid that flows into the porous medium. 

The thermophysical parameters of ${\mathrm{SiO}}_{2}$, ${\mathrm{Al}}_{2}O_{3}$, and CuO particles are shown in Table 6. By using the numerical values of these thermophysical parameters and the nanofluid empirical equations in Table 7, the nanofluid characteristics were extracted, and the effects of adding various nanoparticles on the physical characteristics were examined. Furthermore, for all three nanofluids, the influence of a volume fraction ranging from 1% to 5% on the thermophysical characteristics of the base fluid was investigated. 

The impact of the volume fraction on the density of water nanofluids ${\mathrm{SiO}}_{2}$, ${\mathrm{Al}}_{2}O_{3}$, and CuO is shown in Figure 4a. The density of all three types of nanofluids increased as the volume fraction increased, with the increase being greater for CuO water nanofluids than ${\mathrm{SiO}}_{2}$ water and ${\mathrm{Al}}_{2}O_{3}$ nanofluids. Due to the lowest density of ${\mathrm{SiO}}_{2}$ nanoparticles compared to the others, ${\mathrm{SiO}}_{2}$ water showed the lowest nanofluid density.

Figure 4b depicts the effect of increasing the amount of nanoparticles in the base fluid on thermal capacity. It demonstrates a reduction in thermal capacity as the ∅ value increases. The analysis of the three nanofluid combinations revealed that ${\mathrm{SiO}}_{2}$ in the base water had the maximum thermal capacity. The thermal capacity varies depending on the nanofluid density and the nanoparticles’ thermal capacity. 

Figure 5a demonstrates how increasing the concentration of nanoparticles ∅ influences the nanofluid’s viscosity. Using the viscosity equation (see Table 7), it is clear that the viscosity was only impacted by the parameter ∅ and viscosity of the base fluid and was unaffected by the kind of nanoparticle. The viscosity increased with increasing concentration for all three nanoparticles. Figure 5b shows the impact of increasing the concentration of nanoparticles on the nanofluid’s thermal conductivity. As the concentration of nanoparticles increases, so does the heat conductivity. The greatest thermal conductivity was found in an ${\mathrm{Al}}_{2}O_{3}$ water nanofluid, followed by ${\mathrm{SiO}}_{2}$ and CuO.

The fluid characteristics were a mixture of nanofluids and oils, and their saturation must also be considered in the case of heat transmission. Table 8 shows the thermophysical properties of the energy equation. 

The reservoir rock characteristics were required in the energy calculation for a porous medium. The Rafaat et al. [73] model was adopted because the relative permeability requires the diameter of the granules inside the reservoir; these characteristics are provided in Table 9.

### 2.6. Verification

The simulation was compared to the findings of an experimental study provided by Maghzi et al. [78] to validate the numerical solver. In their experiment, the silica nanoparticles in distilled water were employed to increase the amount of oil recovered from the porous medium. The geometry of their model featured a 2D porous medium with porosity of 0.33, absolute permeability of 200 mD, and size of 6×6 ${\mathrm{cm}}^{2}$. In this work, the results are provided utilizing the COMSOL solver. In Figure 6, the experimental data are compared to the model findings for changes in the amount of oil retrieved from the medium versus pore volume and exhibited a good agreement with experimental findings [93].

## 3. Results

The solution domain and porous medium were considered homogenous in this study. Our research focus was the impacts of nanofluids in water flooding applications. The simulations were run twice, once for water and again for nanofluid inlet flow. As previously stated, empirical research has shown that this technique significantly impacts petroleum outflow. The purpose of this study was to validate and analyze its relevance and impact on EOR. The problem was initially addressed with a concentration of 0 (purified water) and then by contemplating the 5% nanoparticle concentration with water, assuming silicone nanoparticles and water fluid usage.

As demonstrated in Figure 7a–c, the quantity of nanofluid saturation in the environment was significantly larger and the extracted oil from porous media was boosted. In Figure 7a–c, the red contour depicts the porous medium region where the trapped oil was replaced by the nanofluid, and the oil was extracted. The blue area of the contour refers to a porous medium region that held oil that had not yet been released.

As demonstrated in Figure 8a–c, the quantity of oil saturation in the environment was significantly lower and the nanofluid inserted into porous media was boosted. In Figure 8a–c, the blue contour depicts the porous medium region where the trapped oil was replaced by the nanofluid, and the oil was extracted. The red area of the contour refers to a porous medium region that has held oil that has not yet been released.

As seen in Figure 9a–c, the quantity of pressure in the porous medium did not change significantly as a result of pressure varying with velocity. Since Darcy’s law is only applicable for slow flow, most groundwater flow cases fall into this category. In this investigation, the velocity $u_{\mathrm{nf}0}$ was 0.001 m/s, implying a Reynolds number smaller than unity, indicating that the flow was laminar, and that Darcy’s equation was used. 

Figure 10 presents the oil recovery factors for flooding of water and nanofluids. This figure shows that flooding by nanofluids was more effective than flooding by water. As demonstrated in Figure 10, adding nanoparticles to the base fluid improved the recovery factor. The addition of nanoparticles to the base fluid increased the relative weight of the input fluid, increasing energy and allowing for more efficient oil production. Furthermore, because of the nanoparticles’ strong thermal transfer characteristics, the heat was transferred to the oil inside the reservoir more quickly. Oil recovery was improved due to the decreased viscosity and density of reservoir oil. 

Another critical parameter in EOR and the solution of two-phase flows is the relative permeability of the two fluids. In the current study, the relative permeability of water and oil fluid was altered when nanoparticles were added to the water. As demonstrated in Figure 11, raising the relative permeability of nanofluid while simultaneously reducing the permeability of oil due to the presence of nanoparticles were the primary reasons for EOR. Because the relative permeability of the water was enhanced as a result of this procedure, it drove the oil flow better. In conclusion, employing nanofluids instead of water increased the performance of the EOR process. 

There are several ways nanofluids may be used with nanoparticles, and they are dependent on the purposes. Nanoparticles with smaller diameters and higher densities are preferred over other particles in the EOR process. When the input fluid reached the reservoir at a high temperature, the thermal characteristics of the nanoparticles were also considered. According to the characteristics indicated, the three widely utilized nanoparticles in the procedure were ${\mathrm{SiO}}_{2}$, ${\mathrm{Al}}_{2}O_{3}$, and CuO. As seen in Figure 12, these three nanoparticles have only a different impact on the recovery factor of EOR. However, the recovery rate of silicon nanoparticles is approximately 2% higher than that of the other nanoparticles.

By using nanoparticles in the fluid, the specific heat capacity of the fluid decreased, and the heat transfer coefficient of the fluid was exceptionally high. Due to the ease and speed of heat transfer input temperature, in this case, rapid changes and increased process performance were possible. The recovery coefficient was determined for two different fluid input temperatures in Figure 13. The amount of oil produced from the reservoir increased dramatically when the fluid intake temperature rose. Furthermore, because of the nanoparticles’ strong thermal transfer characteristics, the heat was transferred to the oil inside the reservoir more quickly. As a result of the decreased viscosity and density of reservoir oil, the recovery factor was improved.

## 4. Conclusions

To simulate nanofluid and oil reservoir flooding, a two-dimensional rectangular porous homogenous area was filled with oil to simulate nanofluid and oil reservoir flooding. The effect of nanoparticle volume fraction on flooding and how it varied from water flooding was investigated. The EOR process was also simulated and compared with three types of nanoparticles: ${\mathrm{SiO}}_{2}$, ${\mathrm{Al}}_{2}O_{3}$, and CuO. Studies were conducted to determine how the fluid intake temperatures affect the recovery factor, leading to the following conclusions:
Adding 5% silicon nanoparticles to a base fluid in a porous homogenous medium enhanced the oil recovery by more than 20%.Even though the difference in the effects of the three nanoparticles described was less than 2%, silicon had a 2% better oil recovery rate than ${\mathrm{Al}}_{2}O_{3}$ and CuO.Using two different temperatures to calculate the EOR recovery coefficient for heterogeneous porous geometry of 300 and 350 K revealed that temperature was crucial in the EOR process due to its influence on oil viscosity and density decrease. This impact was amplified when utilizing nanofluids for heat transmission due to the efficient and rapid heat transfer.Raising the relative permeability of nanofluid while simultaneously reducing the relative permeability of oil due to the presence of nanoparticles was the primary reason for EOR.The recovery factor of silicon nanoparticles is higher than the recovery factors of aluminum oxide and copper oxide nanoparticles.Using the empirical correlations of the nanofluid thermophysical properties on the relative permeability equations of the nanofluid and oil in the proposed model, it is more accurate than other models to determine the higher recovery factor of one nanoparticle compared to other nanoparticles.

## Figures and Tables

**Figure 1 nanomaterials-12-01011-f001:**
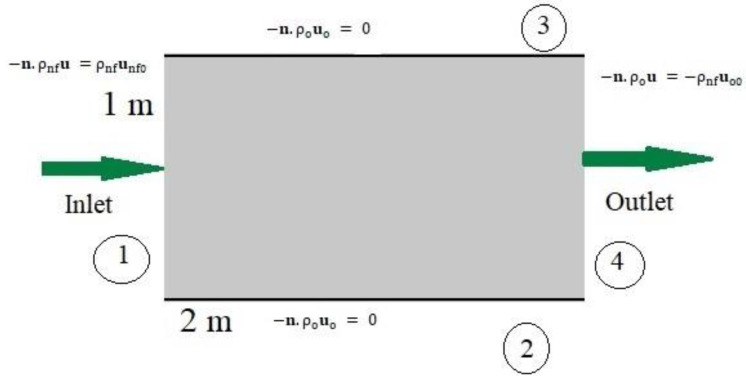
The planned pattern’s schematic geometry.

**Figure 2 nanomaterials-12-01011-f002:**
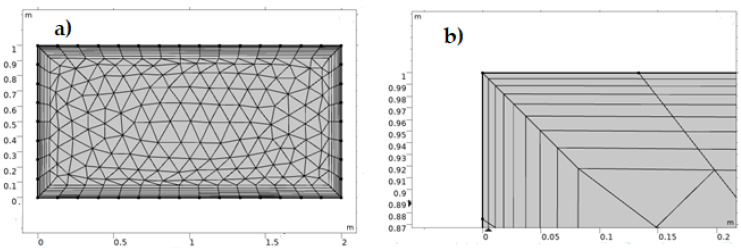
(**a**) Mesh geometry using the boundary layer mesh and (**b**) illustration of the actual structure of the boundary layer mesh.

**Figure 3 nanomaterials-12-01011-f003:**
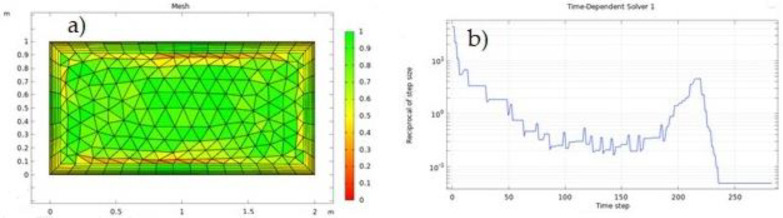
(**a**) Inspection of the geometry mesh in COMSOL Multiphysics and (**b**) convergence plot.

**Figure 4 nanomaterials-12-01011-f004:**
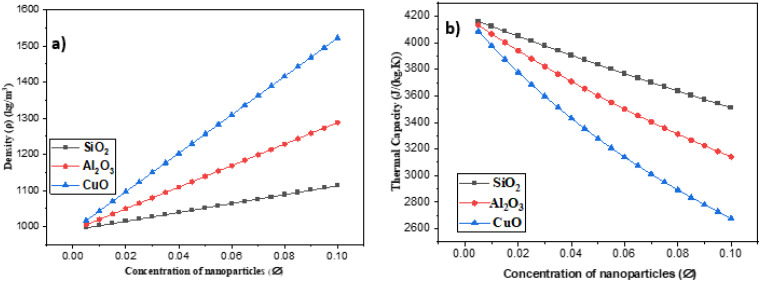
(**a**) The influence of nanoparticles on density and (**b**) nanoparticles’ influence on thermal capacity.

**Figure 5 nanomaterials-12-01011-f005:**
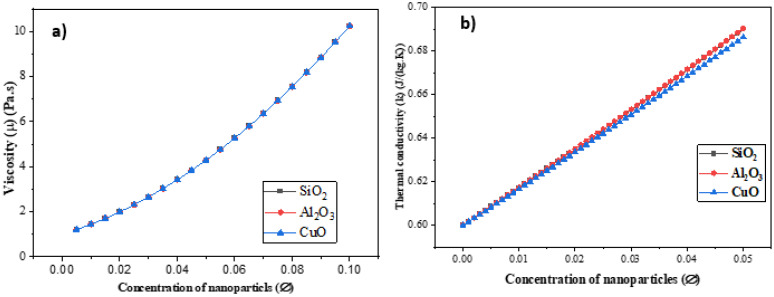
(**a**) Influence of nanoparticles on viscosity and (**b**) effect of nanoparticles on thermal conductivity.

**Figure 6 nanomaterials-12-01011-f006:**
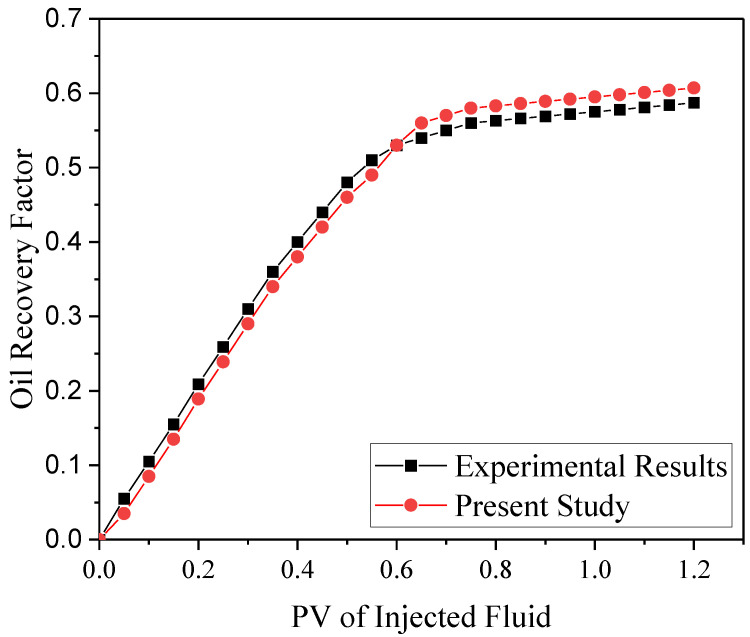
Evaluation of curve changes in the amount of oil recovered from porous medium computed from the model and experimental data [93].

**Figure 7 nanomaterials-12-01011-f007:**
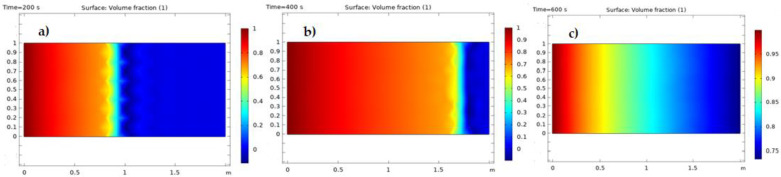
(**a**) Saturation of the nanofluid at 200 s, (**b**) saturation of the nanofluid at 400 s, and (**c**) saturation of the nanofluid at 600 s.

**Figure 8 nanomaterials-12-01011-f008:**
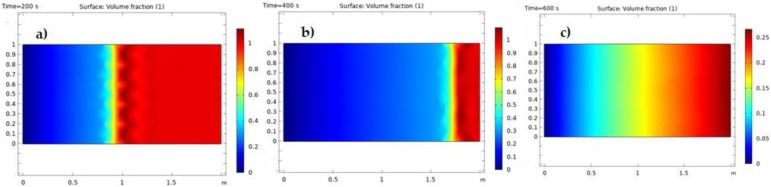
(**a**) Saturation of the oil at 200 s, (**b**) saturation of the oil at 400 s, and (**c**) saturation of the oil at 600 s.

**Figure 9 nanomaterials-12-01011-f009:**
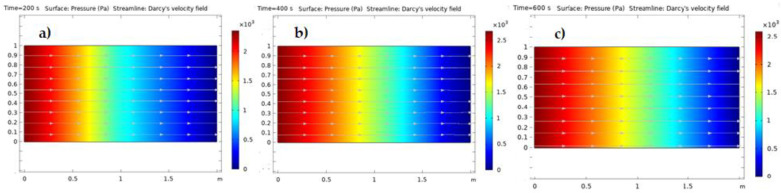
(**a**) Pressure distribution inside the porous medium at 200 s, (**b**) pressure distribution inside the porous medium at 400 s, and (**c**) pressure distribution inside the porous medium at 600 s.

**Figure 10 nanomaterials-12-01011-f010:**
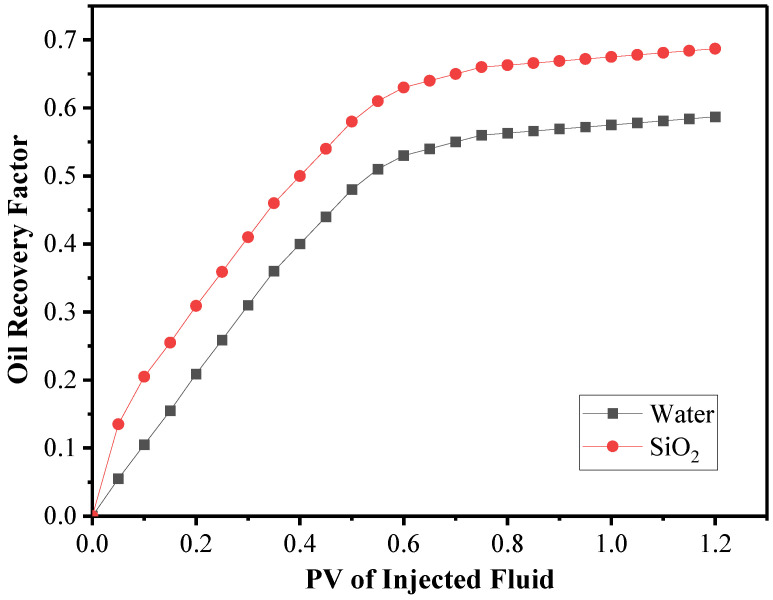
Comparison of the water and the nanofluid recovery factor.

**Figure 11 nanomaterials-12-01011-f011:**
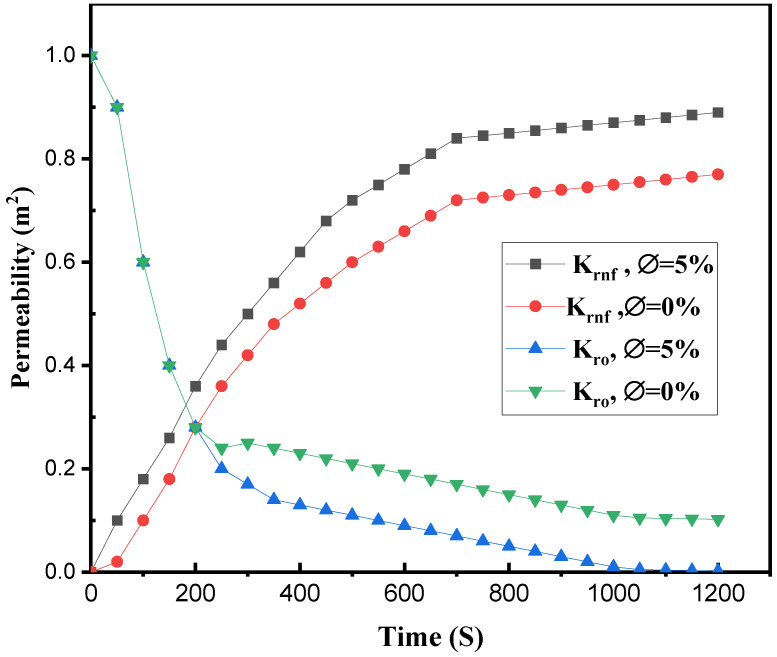
Comparing water/oil relative permeability with null and 5% volumetric nanoparticles in the homogenous porous medium.

**Figure 12 nanomaterials-12-01011-f012:**
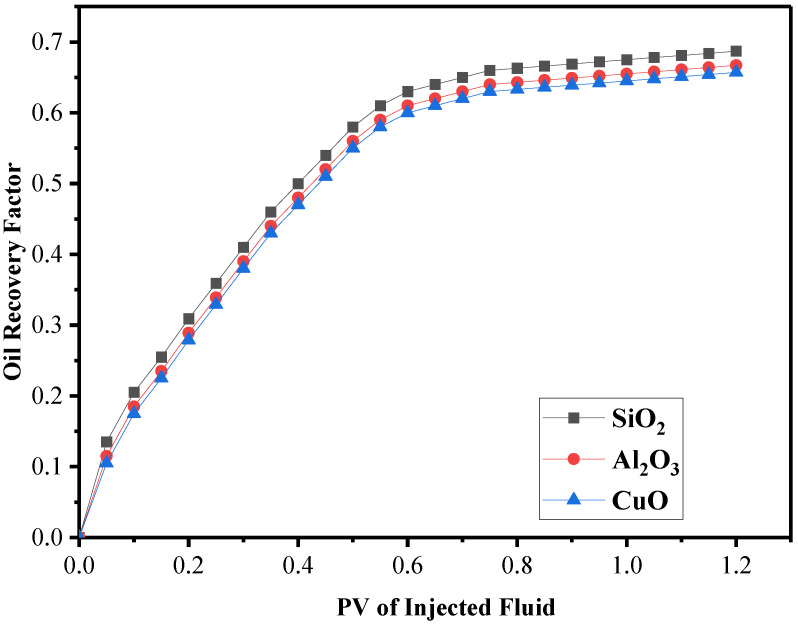
Comparison of nanoparticles SiO_3_, Al_2_O_3_, and CuO recovery coefficients.

**Figure 13 nanomaterials-12-01011-f013:**
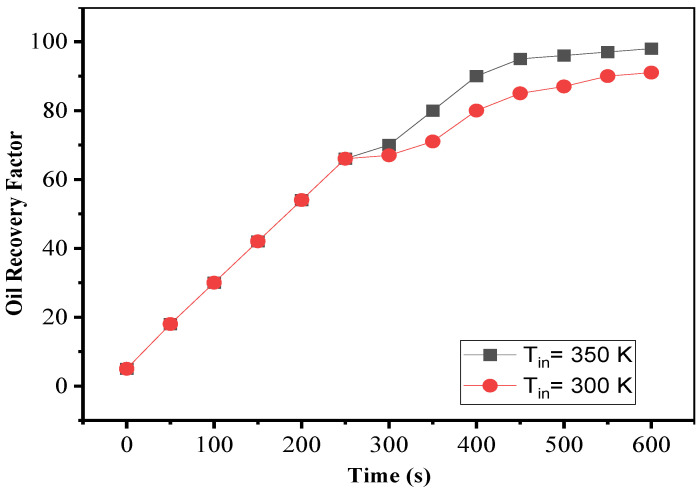
The effect of inlet fluid temperature in a porous homogenous medium.

**Table 1 nanomaterials-12-01011-t001:** Pressure drops in model at different grids.

No.	No. of Elements	∆P (Pa.)
1	125,000	0.875
2	140,000	0.926
3	250,000	0.932
4	300,000	0.933
5	400,000	0.934

**Table 2 nanomaterials-12-01011-t002:** The characteristics of water and oil at a constant temperature.

Properties	Water [80]	Oil [81]	Unit
	Value	Value	
Density (ρ)	990	880	$\frac{\mathrm{kg}}{m^{3}}$
Viscosity (μ)	0.001	4.5e−4	Pa.s
$\mathrm{Specific}\;\mathrm{heat}\;\mathrm{capacity}\;\left(C_{P}\right)$	4200	1670	$\frac{J}{\mathrm{kg}.K}$
$\mathrm{Thermal}\;\mathrm{conductivity}\;\mathrm{coefficient}\;(k_{eff}$)	0.6	0.13	$\frac{W}{m.K}$

**Table 3 nanomaterials-12-01011-t003:** Effects of temperature on viscosity of water [79].

Temperature Efficiency	Viscosity Equation
273.15<T<413.15	$\mu \left(T\right)=1.3799566804-0.021\;*\;T+1.36045e\;–\;4*\;T^{2}-4.6454090e-7\;*\;T^{3}+8.9042735e-10\;*\;T^{4}-9.079069e-13\;*\;T^{5}+3.845733e-16\;*\;T^{6}$
413.15<T<553.75	$\mu \left(T\right)=0.004012-2.1074e-5\;*\;T+3.85772275e\;–\;8\;*\;T{\;}^{2}-2.39730284e-11\;*\;T^{3}$

**Table 4 nanomaterials-12-01011-t004:** Temperature-related changes in oil density [82].

Temperature (°C)	$\mathrm{Value}\;\mathrm{of}\;\mathrm{Oil}\;\mathrm{Density}\;\left(\frac{\mathrm{kg}}{m^{3}}\right)$
15	837.4
20	833.8
40	819

**Table 5 nanomaterials-12-01011-t005:** Temperature-related changes in oil viscosity [82].

Temperature (°C)	Value of Oil Viscosity (Pa·s)
15	0.01692
20	0.01384
40	0.006969

**Table 6 nanomaterials-12-01011-t006:** Nanoparticle properties.

Properties	$SiO_{2}$ [83]	$Al_{2}O_{3}$ [84]	CuO [85]	Unit
Density (ρ)	2220	3970	6310	$\frac{\mathrm{kg}}{m^{3}}$
Molecular weight	60	101.96	79.55	$\frac{g}{\mathrm{mol}}$
Specific heat capacity (CP)	745	765	531	$\frac{J}{\mathrm{kg}.K}$
Thermal conductivity coefficient (K)	36	40	20	$\frac{W}{m.K}$
Diameter	40	30	30	nm

**Table 7 nanomaterials-12-01011-t007:** Properties of nanofluids.

Nanofluid Properties	Equation (Combining Nanoparticle and Water)	Unit
$\mathrm{Density}\;\left(\rho_{\mathrm{nf}}\right)$ [86]	$\rho_{\mathrm{nf}}=\emptyset \rho_{np}+\left(1-\emptyset \right)\rho_{w}$	$\frac{\mathrm{kg}}{m^{3}}$
$\mathrm{Specific}\;\mathrm{heat}\;\mathrm{capacity}\;(C_{P_{\mathrm{nf}}}$) [87]	$C_{P_{\mathrm{nf}}}=\frac{\emptyset \rho_{np}C_{Pnp}+\left(1-\emptyset \right)\rho_{w}C_{Pw}}{\rho_{nf}}$	$\frac{J}{\mathrm{kg}.K}$
$\mathrm{Thermal}\;\mathrm{conductivity}\;\mathrm{coefficient}\;(K_{\mathrm{nf}}$) [88,89]	$K_{\mathrm{nf}}=k_{w}\frac{k_{np}+2k_{w}-2\emptyset \left(k_{w}-k_{np}\right)}{k_{np}+2k_{w}+\emptyset \left(k_{w}-k_{np}\right)}$	$\frac{W}{m.K}$
$\mathrm{Viscosity}\;(\mu_{\mathrm{nf}}$) [90,91]	$\mu_{\mathrm{nf}}=\mu_{w}\left(1+39.11\emptyset +533.9\emptyset^{2}\right)$	Pa.s

**Table 8 nanomaterials-12-01011-t008:** Total characteristics that may be used in the energy equation.

Nanofluid Properties	Equation (Combining Nanoparticle and Water)	Unit
$\mathrm{Density}\;(\rho_{\mathrm{tot}}$)	$\rho_{\mathrm{tot}}=S_{\mathrm{nf}}\rho_{\mathrm{nf}}+\left(1-S_{\mathrm{nf}}\right)\rho_{o}$	$\frac{\mathrm{kg}}{m^{3}}$
$\mathrm{Specific}\;\mathrm{heat}\;\mathrm{capacity}\;\left({\mathrm{Cp}}_{\mathrm{tot}}\right)$	${\mathrm{Cp}}_{\mathrm{tot}}=S_{\mathrm{nf}}{\mathrm{Cp}}_{\mathrm{nf}}+\left(1-S_{\mathrm{nf}}\right){\mathrm{Cp}}_{o}$	$\frac{J}{\mathrm{kg}.K}$
$\mathrm{Thermal}\;\mathrm{conductivity}\;\mathrm{coefficient}\;\left(K_{\mathrm{tot}}\right)$	$K_{\mathrm{tot}}=S_{\mathrm{nf}}K_{\mathrm{nf}}+\left(1-S_{\mathrm{nf}}\right)k_{o}$	$\frac{W}{m.K}$

**Table 9 nanomaterials-12-01011-t009:** Properties of reservoir rocks [92].

Properties of Reservoir Rocks	Value	Unit
Density (ρ)	2714	$\frac{\mathrm{kg}}{m^{3}}$
$\mathrm{Specific}\;\mathrm{heat}\;\mathrm{capacity}\;\left(C_{P}\right)$	851	$\frac{J}{\mathrm{kg}.K}$
$\mathrm{Thermal}\;\mathrm{conductivity}\;\mathrm{coefficient}\;\left(k_{eff}\right)$	2.2	$\frac{W}{m.K}$
Diameter of particle	3	nm

## Data Availability

The data presented in this study are available from the corresponding author upon reasonable request.

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
