# Peer review of "Thermophysical Properties of Nanofluid in Two-Phase Fluid Flow through a Porous Rectangular Medium for Enhanced Oil Recovery"

_nanomaterials, 2022, doi:10.3390/nano12061011_

Round 1

Reviewer 1 Report

This paper studied the effect of different nanoparticle volume fraction on flooding. The novelty of present study should be clearly indicated. The following comments can be considered to make a revision of present version.

  1. The novelty of present study should be clearly indicated in the abstract and in the introduction.
  2. The authors stated that ‘there is a boundary layer right along with the contact between the nanofluid and the solid as shown in Figure 2.’ But there is no information about fluid and the solid in figure 2. Please clearly present this information in Figure 2.
  3. In figure 2 about the mesh, it seems that there is no difference between the boundary layer and other region. Mesh is not refined in the boundary layer?
  4. Provide information about the grid independence study.
  5. Have the results evaluated by the experiment or results in the literature?
  6. It seems that Fig. 19 is not discussed in the paper.
  7. There are too many figures, some figures can combined together as one figure.
  8. Deep discussion are needed for the results.
  9. Please provide quantitative results in both the discussion and in the conclusions.

Author Response

Response to Reviewer 1 Comments

The authors would like to thank the reviewer for the valuable comments and suggestions that contribute to improving our manuscript. Amendments were provided accordingly, and all modifications were highlighted red with track changes in the revised version of the manuscript. Response to the comments is given below.

Point 1: The novelty of the present study should be clearly indicated in the abstract and in the introduction.

Response 1: Thank you for your valuable comment. The novelty of study has been indicated in the abstract and in the introduction (please see lines 24-28 in the abstract and lines 91-105 in the introduction).

Point 2: The authors stated that there is a boundary layer right along with the contact between the nanofluid and the solid as shown in Figure 2. But there is no information about fluid and the solid in figure 2. Please clearly present this information in Figure 2.

Response 2: Thank you . We have included the information and added figure 2a and figure 2b as suggested (please see the added figure 2 and information added in lines 142-146 in the revised manuscript).

Point 3: In figure 2 about the mesh, it seems that there is no difference between the boundary layer and other region. Mesh is not refined in the boundary layer?

Response 3: Thank you . Mesh is redrawn as shown in Figure 2 but the previous figure was exported from COMSOL Multiphysics software before using the boundary layer mesh.(please see the revised maunscript).   

Point 4: Provide information about the grid independence study

Response 4: Thank you. Grid independency details and information have been added in the revised manuscript (please see lines 152-167) and you may also see the information added in Table 1 and figure 3.

Point 5: Have the results evaluated by the experiment or results in the literature?

Response 5:  Thank you . The results were evaluated by the results in the literature.

Point 6: It seems that Fig. 19 is not discussed in the paper.

Response 6: Thank you . The necessary information about the Fig. 12 has been added in the revised manuscript.

Point 7: There are too many figures, some figures can combine together as one figure.

Response 7: Thank you . Many figures have been redrawn and combined together as your valuable suggestion.

Point 8: Deep discussion are needed for the results.

Response 8: Thank you. The necessary information has been added.

Point 9: Please provide quantitative results in both the discussion and in the conclusions.

Response 9: Thank you for your valuable comment. The quantitative reuslt and findings have been added in the conclusion.(please see lines 453-461 in the revised manuscript).

Reviewer 2 Report

This study investigated oil extraction in a two-phase incompressible fluid in a two-dimensional rectangular porous homogenous area filled with oil and having no capillary pressure. Three nanoparticles including SiO2, Al2O3, CuO are introduced to compare oil recovery. This paper can be published in nanomaterials

  • Figure images quality is not good. Should be improved.
  • Why you selected the SiO2, Al2O3, CuO among a lot of materials as nanodot? This paper does not show material part strongly. More focused on mathmathics.
  • What is the m located at right-bottom conner in Figure 2
  • What is the unit of x-axis in Figure 4?
  • In Figure 6. Subscription of SiO2….. and phi is just unit. Need carefully check all figures.

Author Response

Response to Reviewer 2 Comments

The authors would like to thank the reviewer for the valuable comments and suggestions that contribute to improving our manuscript. Amendments were provided accordingly, and all modifications were highlighted in red track change in the revised version of the manuscript. Response to the comments is given below.

Point 1: Figure images quality is not good. Should be improved.

Response 1: Thank you. Figure images have been improved. (please see the whole manuscript).

Point 2: Why did you select the SiO2, Al2O3, CuO among a lot of materials as nanodot? This paper does not show material part strongly. More focused on mathmathics.

Response 2: Thank you . Reasons for the selection of silicon oxides , aluminum oxide, and copper oxide among other nanomaterials have been briefly explained in the introduction(please see lines 56-65 in the revised manuscript).

Point 3: What is the m located at the right-bottom conner in Figure 2.

Response 3: Thank you. The meter m is the unit of distance. The geometry of the model was produced using COMSOL Multiphysics software is a rectangular size   .

Point 4: In Figure 6. Subscription of SiO2….. and phi is just unit. Need carefully check all figures.

Response 4: Thank you. Figures have been redrawn and improved(please see the whole manuscript).

Round 2

Reviewer 1 Report

the revised version can be considered for acceptance.